# Riverbank Stability Assessment under River Water Level Changes and Hydraulic Erosion

**Toan Duong Thi * and Duc Do Minh**

VNU University of Science, Vietnam National University, Hanoi, Hanoi 100000, Vietnam; ducdm@vnu.edu.vn
* Correspondence: duongtoan@hus.edu.vn; Tel.: +84-093-454-3261

**Abstract:** The dominant mechanism of riverbank cantilever failure is soil erosion of the bank toe and near bank zone. This paper demonstrates that the shape of the riverbank cantilever failure depends on the properties of the soil and the fluctuation of the river water level (RWL). With a stable RWL, a riverbank with higher resistance force leads to failure with larger and deeper overhang erosion width. When RWL rises, a less cohesive soil bank will be eroded over a larger width and riverbank failure will occur earlier. With a low rate of rising RWL, riverbank failure may happen in a type of mass failure. With a high rate of rising RWL, a riverbank will fail in a type of overhang riverbank failure, with the soil erosion rate being the main affected factor.

**Keywords:** water shear stress; soil erosion; riverbank cantilever failure

---

## 1. Introduction

Riverbank failure occurs by multiple factors, especially during a rainy season, such as the fluctuation of river water level (RWL), groundwater, pore water pressure, soil strength (soil suction and shear strength), and soil erosion. The two main mechanics of riverbank failure mentioned in most previous research are mass failure and cantilever failure [1]. The role of the related factors in mass failure mechanics have been studied quite clearly in [2–9], which found that the relationship between the soil hydraulic and the rate of RWL change is the main factor controlling these factors as well as the riverbank stability. While the cantilever failure mechanic is more complex, it often includes not only change in pore water pressure and soil strength, but also the process of soil erosion in the toe of the riverbank. To build an understanding of cantilever failure mechanics, a couple of the factors in the mass failure and soil erosion will be analyzed in this paper.

Soil erosion is one of the main processes having a significant impact on riverbank stability and has often been mentioned by hydraulic dynamic erosion [10–33], and seepage erosion [34–43]. Riverbank failure deals with the soil fluvial erosion known as the cantilever type, which is a combination of undercutting erosion and mass collapse of the upper part of the riverbank [10,17,18,20,22]. Soil erosion occurs when the water shear stress caused by water flow around the interface of water and soil is higher than the critical shear stress [44]. Shear stress is the pressure caused by water flow around the interface of water and soil. In the rainy season, the river water level increases and flows with a high shear stress, expanding the soil erosion width and creating an overhanging riverbank shape.

To understand the mechanism of the cantilever riverbank failure, previous research has attempted to combine the processes of soil erosion and change of riverbank geometry into riverbank stability analysis. The Department of Agriculture's Agricultural Research Service developed the Bank-Stability and Toe-Erosion Model (BSTEM), which can be used to estimate the erosion of banks and bank-toe materials [10–16,22]. The GeoSlope program has been used to aid understanding of the mechanisms of riverbank failure coupled with soil bank erosion and riverbank groundwater seepage [5,17,21]. Other models have also been suggested for understanding and analysis of overhanging riverbank failure.

Yong et al. (2014) [22] used the extensively verified and validated 2D depth-averaged mobile-bed model SRH-2D and the BSTEM to predict channel adjustment and platform development. This was considered the first study to accomplish a complete coupling of a multilayer cohesive bank model and a multidimensional mobile-bed model. Patsinghasanee et al. (2015, 2017, 2018) [24,25,30] developed a coupled model implementing a triple-grid approach from/to a Cartesian grid cell, consisting of a coarse one-dimensional (1D) grid for the flow field in the lateral direction; a fine 1D grid for sediment transport, and bed deformation in the lateral direction; and a 2D grid for cantilever failure in the vertical and lateral directions. Abderrezzak (2016) [28] focused on the simulation of non-cohesive, non-uniform bank material, implemented a simple bank failure in SISYPHE and the sediment transport module of TELEMAC-MASCARET. From those researches it can be seen that fluvial erosion has significant impact on riverbank stability. In addition, the mechanics of cantilever riverbank failure give rise to complex problems, including instantaneous processes such as soil erosion, change of bank geometry, groundwater level, and unsaturated soil properties. These processes all require numerical analysis, for which previous models are not adequate. Analysis of the cantilever riverbank requires further development.

Regarding the factors affecting cantilever riverbank failure, the physical properties of the soil, particularly grain size and density, are the main factors affecting both the fluvial erosion rate and riverbank stability [19,20,23–25,30]. Samadi et al. (2011) [19] performed the first experiment involving the creation of an overhanging riverbank block to understand the failure mechanism of overhanging blocks. The experimental results were compared with the results obtained from a numerically simulated model [20]. By analyzing riverbank soils with different densities, Samadi et al. (2011, 2013) [19,20] concluded that with increasing soil density, the undermining depth at which failure occurs will increase, as will the width ratio of upper to lower overhanging blocks in failure conditions. Yu et al. (2015) [23] showed different mechanics of riverbank failure owing to different soil materials. For cohesive soil, as the sediment-moving incipient velocity is much greater than that of non-cohesive soil and its bed erosion is weaker, bank failure occurs deeper, near the bank toe. The non-cohesive bank tends to collapse near the water surface. Patsinghasanee et al. (2017, 2018) [24,25] recommended that the physical and erosion properties of soil should continue to be investigated in further studies.

From an overview of the previous research focused on soil erosion and riverbank failure by water flow stress, it may appear that soil properties are important factors impacting on soil erosion and riverbank failure [19,20,23–25,30,43–56]. However, few studies have focused on the effects of soil properties and erosion on riverbank cantilever failure. Moreover, analysis of riverbank erosion should include the effects of the river water level (RWL), which changes during the flooding season due to heavy rains and sudden recharge from upstream recharge. In most previous studies, the RWL has been held constant, and the pore pressure and groundwater were not accounted for. The primary objectives in this paper as follows. (i) Analysis of the effect of soil erosion on riverbank failure. Three different kinds of bank soils were analyzed in cases both with erosion and without soil undercutting erosion. (ii) Discussion of the mechanics of riverbank failure, including the effect of factors such as soil property and river water level change. The discussion of these and previous results will improve our understanding of riverbank failure mechanics.

## 2. Methods

Figure 1 shows the framework of the present study, which includes two main parts, namely, a laboratory soil experiment and simulation models in the GeoSlope program. Laboratory soil properties were soil erosion, saturated soil properties (i.e., physical properties, shear strength, and hydraulic conductivity), and unsaturated soil property (soil suction).

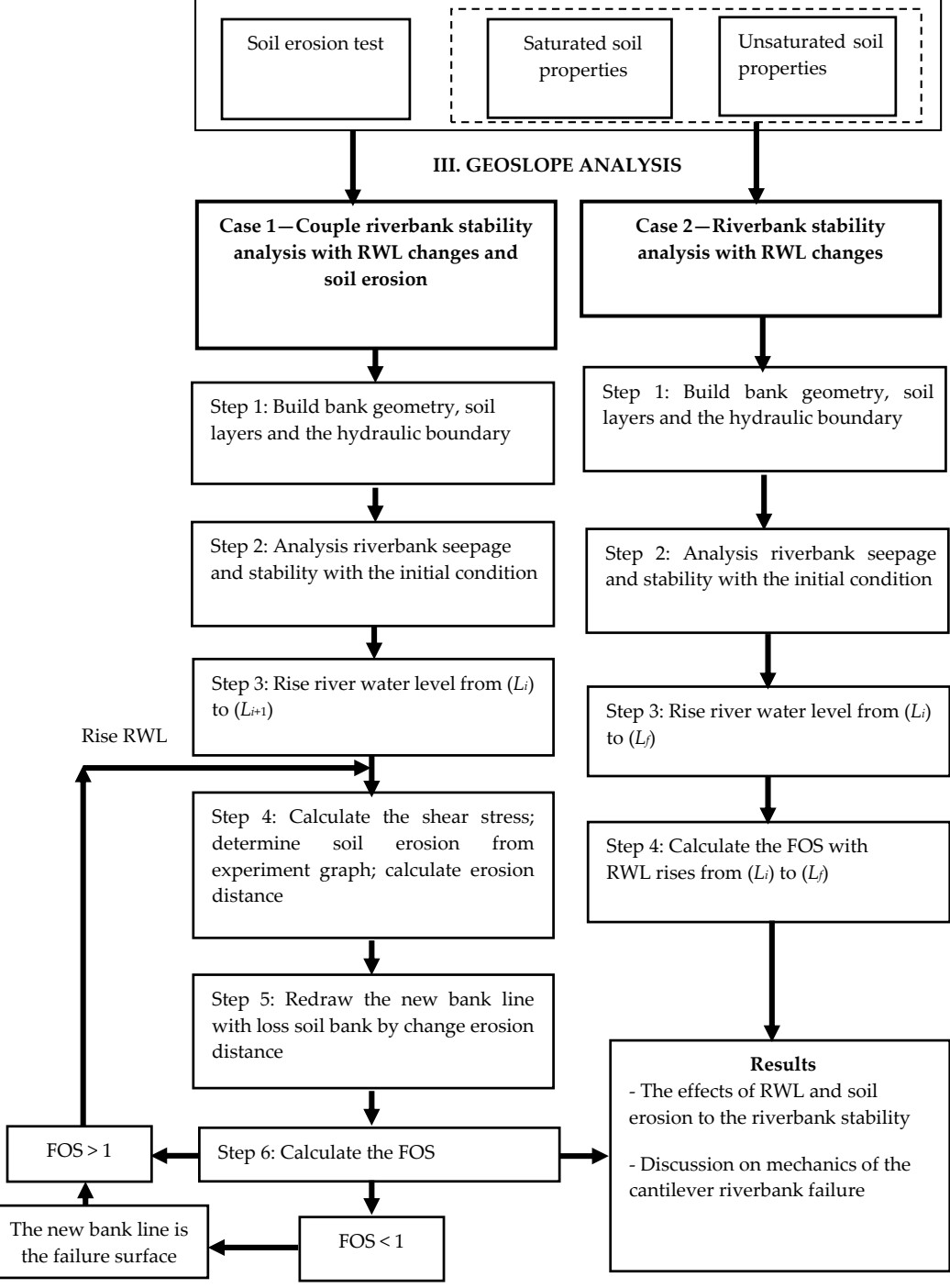

**Figure 1.** The research framework.

The physical properties, shear strength, and hydraulic conductivity were determined following the standards of ASTM D 2216, D 2937-00, JGS 0524: 2000, and JIS A: 1218, respectively. Soil suction was determined using a pressure plate apparatus, as shown in [57]. A description of the equipment and method used to determine soil erosion velocity is provided below. This research used the GeoSlope program to analyze riverbank stability in two cases. In the first case, riverbank stability was analyzed coupled with soil erosion and change of river water level. In this, all three soil laboratory experiment

groups (soil erosion, and saturated and unsaturated soil properties) were used in the analysis process. In the second case, riverbank stability was analyzed with only river water level change, and without soil bank erosion. The details of the soil erosion test and analysis model in the GeoSlope program are described below.

### 2.1. Determining Soil Erosion Rate in Laboratory

Soil erosion, which in the bank toe is caused by river water flow, was determined in an open pipe in the laboratory. This test was based on the erosion function apparatus invented by Briaud et al. (1999, 2001a, 2001b, 2004, 2008) [43,53–56]. The equipment includes an open rectangular pipe 2 m long, 0.2 m wide, and 0.1 m high (Figure 2). The water flowing through the pipe is supplied by a water tank. The slope of the water pipe and the height of the water tank can be changed to control the water velocity. A cylindrical mold attached to the bottom of the water pipe has a diameter of 7.6 cm; this is used to compact the soil sample. The protrusion of soil in this mold can be manipulated by hand control in the bottom of the soil column. The experimental model measures the time needed to erode a certain soil height by water flowing through a simulated opening channel. The soil erosion rate (mm/h) is thus defined as the ratio of the eroded height (mm) to the elapsed time (h).

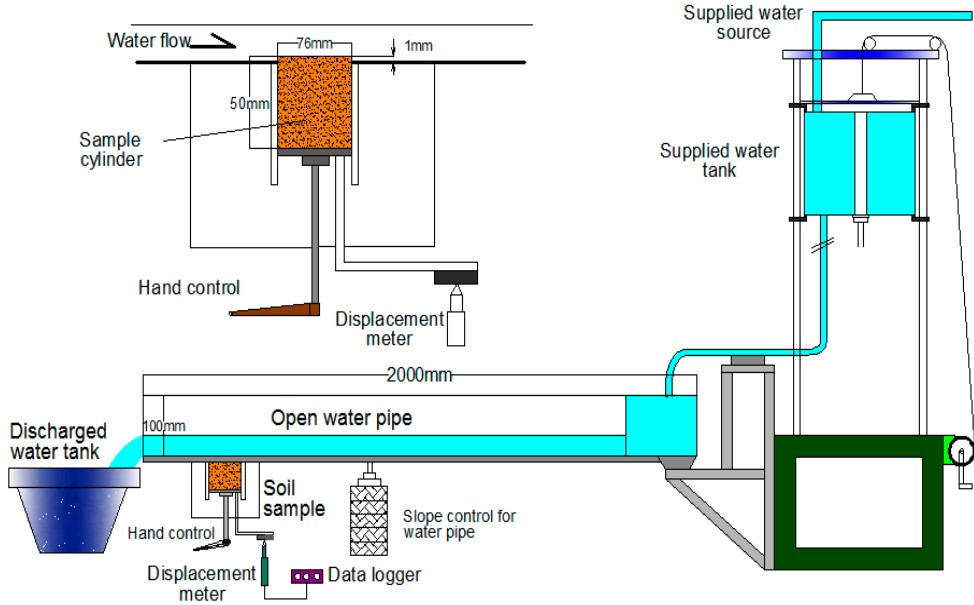

**Figure 2.** Configuration of soil erosion test.

In this analysis, a soil sample with predetermined water content and dry density was mounted in the sample cylinder. During the test, the core soil was made to protrude in intervals of 1 mm higher than the bottom of the water pipe (Figure 3). The velocity of the water flow was set at a predetermined rate (m/s) and continued flowing until 1 mm of soil was completely eroded. The time it took to erode 1 mm was recorded, and this represents the duration $\Delta t$ needed to erode the height of the soil $\Delta h$ (mm). The results of this analysis represent the relationship between the erosion rate and shear stress (Figure 4), in which the erosion rate and shear stress are calculated using the following equations. The soil erosion rate ($\varepsilon$) is the ratio of the height of the eroded soil ($\Delta h$) to the elapsed time ($\Delta t$) [53–56,58–62]:

$$\varepsilon = \Delta h/\Delta t \tag{1}$$

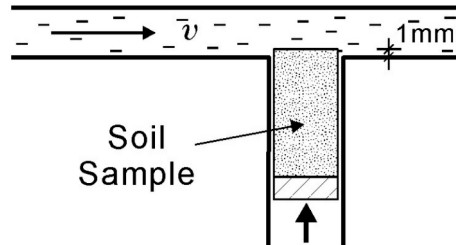

**Figure 3.** Soil sample protruding 1 mm before the start of water flow.

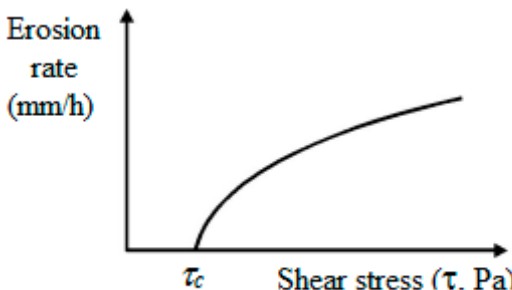

**Figure 4.** Result curve: relationship between soil erosion rate and water shear stress.

The shear stress from the water flow can be calculated using the Moody chart:

$$\tau = \frac{\rho f V^2}{8} \tag{2}$$

in which $\tau$ is the shear stress (N/m$^2$ or Pa), $\rho$ is the density of water, $V$ is the average water velocity, and $f$ is a friction coefficient. Theoretically, $f$ is obtained from the Moody diagram using the relative roughness (roughness height/conduit diameter) and the Reynolds number, which is calculated as:

$$R_e = \frac{VD}{v} \tag{3}$$

where $D$ is the hydraulic diameter of the pipe and $v$ is the kinematic viscosity of water. The hydraulic diameter is calculated as:

$$D = \frac{2ab}{(a+b)} \tag{4}$$

The open pipe is assumed to be smooth; $a$ represents the width, and $b$ represents the height of water flow in the open pipe. For turbulent flow in smooth conduits, the following approximations of the friction coefficient and the Reynolds number dependence can be used. For $R_e > 3000$,

$$\frac{1}{\sqrt{f}} = 2.0 \log(R_e \sqrt{f}) - 0.8 \tag{5}$$

When $R_e < 10^5$, the above equation can be approximated as:

$$f = \frac{0.316}{R_e^{1/4}} \tag{6}$$

Equations (5) and (6) can be used to estimate the friction factor without having to use the Moody chart [58,62], although multiple iterations may be required to obtain convergence. In the curve relating soil erosion rates and water stress, the critical shear stress ($\tau_c$) is defined as the stress value at which

soil begins to erode. Finally, the results are presented by the erosion curve as shown in Figure 4, which provides the relationship between the soil erosion rate and the water flow shear stress.

*2.2. Riverbank Erosion Simulation*

2.2.1. The Boundary Shear Stress Causing Soil Erosion

To assess the erosion of soil at the lateral regions of the riverbank and the bank toe prior to riverbank collapse, the river flow shear stress profile must be determined. The basic method for calculating the shear stress boundary is described using the following equation (Chow, 1959) [59]:

$$\tau_o = \rho_w \sin\theta \cdot \frac{A}{P} = \rho_w \sin\theta \cdot R \tag{7}$$

In this equation, the shear stress is calculated based on the hydraulic radius ($R$), which is determined by dividing the cross-sectional area of the flow ($A$) by the wetted perimeter ($P$), in which $\sin\theta$ is the slope of the river. The shear stress also can be calculated based on the mean flow depth (Hickin, 2004) [60], where the following equation is used:

$$\tau_o = \rho_w \sin\theta(d) \tag{8}$$

where $d$ is the shear depth of the water, which represents the distance from the RWL surface to a calculated point. This equation is applied with the assumption that the flow is steady and uniform.

During a flood event, the RWL continually increases, which induces changes in water shear stress with increasing shear depth. Therefore, it is necessary to use a shear stress profile to simulate likely changes to the lateral geometry and depth of the riverbank. Equation (9) shows the distribution of shear stress from the riverbed to the water surface, where $D$ is the current water depth at boundary, $y$ is the elevation above the boundary of a considered section across the channel, and the shear depth d is equal to $(D - y)$. The shear stress boundary ($\tau_b$) at the interface between the river channel and riverbank can be determined using Equation (10) [60]:

$$\tau_y = \rho_w \sin\theta(d) = \rho_w \sin\theta(D - y) \tag{9}$$

$$\tau_b = 0.75\rho_w \sin\theta(D - y) \tag{10}$$

where $\tau_y$ represents the shear stress acting across the bottom of the block of fluid (i.e., it represents the downslope component of the weight of fluid in the block) at a height y above the riverbed. At the riverbed (i.e., $y = 0$), the shear stress is greatest and is referred to as $\tau_o$; this represents the force per unit area acting on the bed that is available to move sediment. $\tau_b$ represents the shear stress near the riverbank and is used to simulate the water stress-induced erosion of soil at the riverbank.

2.2.2. Soil Erosion and Riverbank Erosion Width

When the boundary shear stress ($\tau_b$) at the riverbank is higher than the critical shear stress ($\tau_c$), which is determined by the erosion test, riverbank soil erosion occurs. Based on the erosion curve determined by the erosion test, the erosion rate responding to the shear stress distribution at the riverbank will be defined. The soil erosion width ($\Delta E$) caused by the increase of the RWL and the time elapsed while submerged ($\Delta t$) is calculated by the following equation:

$$\Delta E = \varepsilon\Delta t \tag{11}$$

where the $\varepsilon$ is the soil erosion rate responding to the specific shear stress and determined by the erosion curve. The value of the soil erosion rate and erosion width will be used in the analysis model of riverbank stability presented in Section 2.3. The applied idea for calculating erosion width in analysis riverbank stability here is the same as for models used in [10–18].

### 2.3. Model Analysis of Riverbank Stability with Seepage and Soil Erosion

This model analyzes riverbank stability under the joint effects of seepage and soil erosion with changing RWL. The concept of this model is modified from the analysis procedure performed using GeoSlope in previous studies [17,18,37–39]. In general, the procedure used to model bank-toe erosion and the changing bank line is presented in Case 1 of Figure 1, and is outlined as follows: Set up the initial riverbank configuration in the SEEP/W module in GeoSlope program to analyze coupled shear stress and pore water pressure to simulate RWL changes and water stress (Step 1). Analysis of seepage and stability response to the initial conditions (Step 2).

The following governing differential equation for two-dimensional seepage was used in SEEP/W to determine the changes in pore pressure state and calculate unsaturated soil properties [44]:

$$\frac{\partial}{\partial x}\left(k_x\frac{\partial H_w}{\partial x}\right) + \frac{\partial}{\partial y}\left(k_y\frac{\partial H_w}{\partial y}\right) + Q = \frac{\partial \theta}{\partial t} = m_w\gamma_w\frac{\partial(H_w - y)}{\partial t} \tag{12}$$

where $k_x$ (resp., $k_y$) is the hydraulic conductivity in the $x$-direction (resp., $y$-direction), $Q$ is the applied hydrodynamic boundary, $\theta$ is the volumetric water content, and $t$ is time, $H_w$ is the total hydraulic head. While

$$\partial \theta = m_w\partial u_w = m_w\gamma_w\partial(H_w - y)u_w = \gamma_w(H_w - y). \tag{13}$$

$m_w$ is the slope of the suction curve and depends on the type of soil, $\gamma_w$ is the unit weight of water and $y$ is the elevation, $(u_w)$ is pore water pressure.

Here, the bank is first built using the initial geometry of the bank line ($l_i$) and the initial RWL ($L_i$) defined at the time $t_i$ (Figure 5a) and, then, the hydraulic boundary as the RWL changes from $L_i$ to $L_{i+1}$, which corresponds to moving in time from $t_i$ to $t_{i+1}$ in (Step 3). The simulation is then run to model transient seepage. During flood events, the RWL is simulated as rising at a fixed rate. Bank erosion occurs in submerged areas below the surface of the RWL, based on its water shear stress (Step 4). The change in the riverbank line, which represents the surface between the soil bank and the water river channel, is designated as the lateral eroded distance or the erosion width ($\Delta E_{i+1}$) (Figure 5b), which is the distance the soil is eroded in a specific elapsed time, $\Delta t_{i+1}$, determined using the following equation:

$$\Delta E_{i+1} = \varepsilon\Delta t_{i+1}; \ \Delta t_{i+1} = t_{i+1} - t_i \tag{14}$$

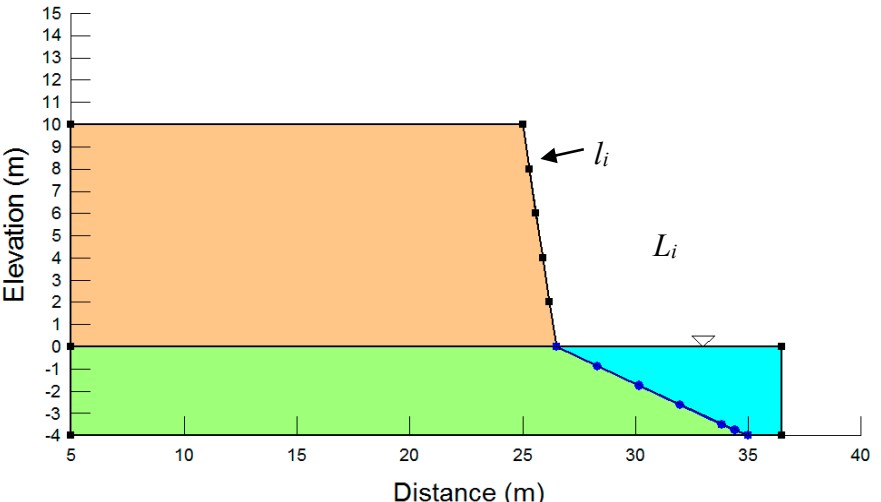

(**a**) Initial geometry of riverbank line ($l_i$) and initial river water level (RWL) ($L_i$)

**Figure 5.** *Cont.*

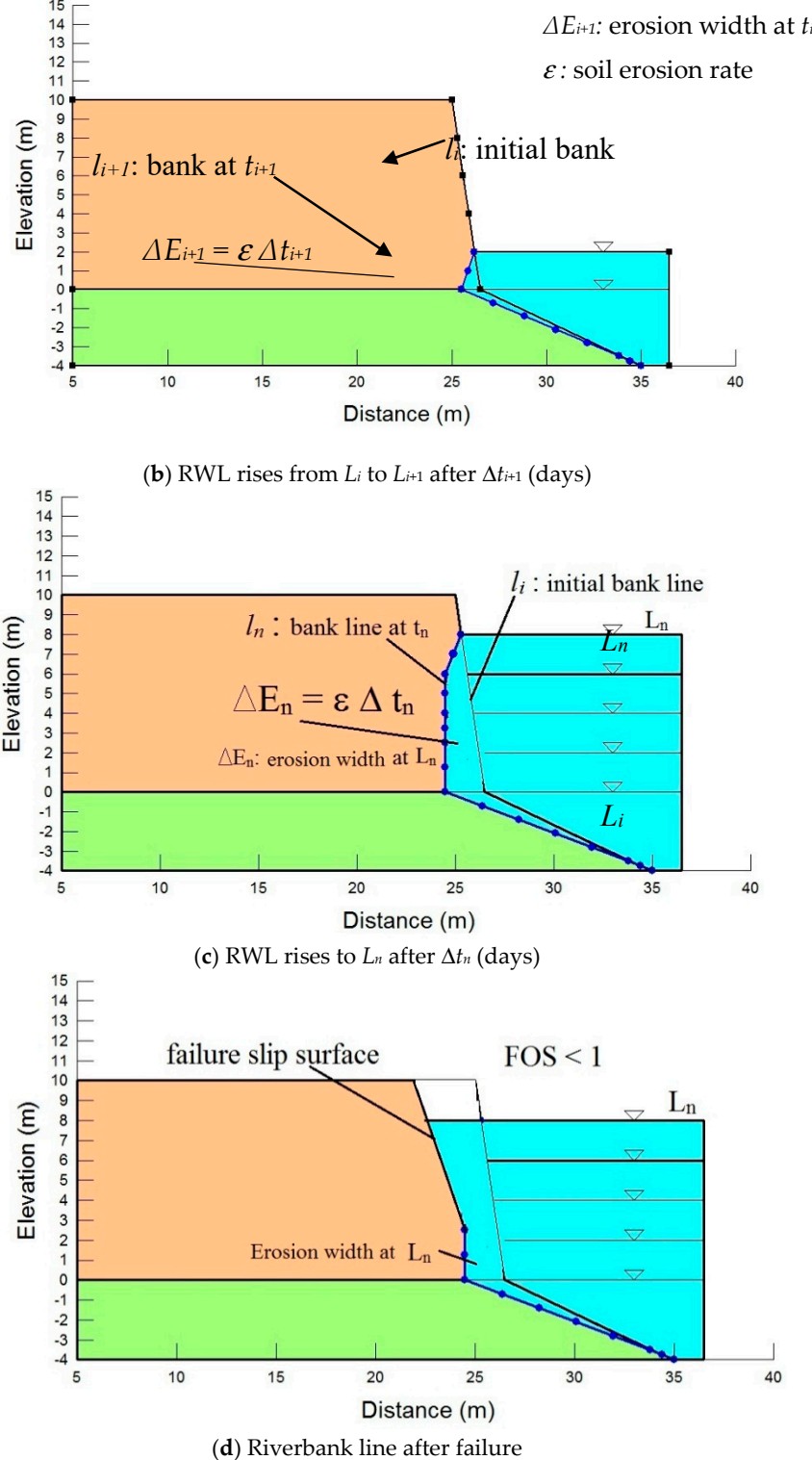

(**b**) RWL rises from $L_i$ to $L_{i+1}$ after $\Delta t_{i+1}$ (days)

(**c**) RWL rises to $L_n$ after $\Delta t_n$ (days)

(**d**) Riverbank line after failure

**Figure 5.** Examples of changes in riverbank geometry due to hydraulic erosion.

In Equation (12), $\varepsilon$ is the soil erosion rate, and $\Delta t$ is the total elapsed time (in days). The soil erosion rate ($\varepsilon$) is determined by erosion tests (as in Section 2.1). At the end of this time period, (i.e., at $t_{i+1}$), the bank line has already been changed by water erosion by a distance $\Delta E_{i+1}$, therefore, a new bank line ($l_{i+1}$) is manually drawn by modifying the point in the lateral bank under the river water

level (Step 5 and Figure 5b). The modify function in the GeoSlope program is used for redrawing the riverbank.

Lastly, the factor of safety (FOS) is calculated by SLOPE/W in the GeoSlope program with the current high river water level, pore water and new bank at which soil under the river water level was eroded (Step 6). In SLOPE/W, it is impossible to simulate the slope of the overhang shape, so the area between the initial bank line ($l_i$) and the new bank line ($l_{i+1}$), was simulated as a low density material with properties similar to those of water. In the finite element modes of SEEP/W, the elements are required to be connected at their corners by nodes that are not representative of an undercutting process, where an element tends to break away from the adjacent element, and the geometry of the riverbank with an overhang cannot be simulated in SLOPE/W [17,18]. To overcome this, the change in geometry is produced by changing the material properties, and the eroded soil area is filled with very low-strength soil.

The obtained FOS is used in further analysis: (1) If the FOS is higher than 1.0, the analysis process returns to Step 3, and RWL will continue to rise to level $L_n$. In this case, the riverbank geometry is continually updated and redrawn by the new lines induced by the development of the erosion width below the current RWL, $\Delta E_n$ (Figure 5c); then recalculate the FOS. (2) If the obtained FOS is less than 1.0, the riverbank fails. The new bank line is updated by both the new erosion width and the failed slip surface (Figure 5d), then the FOS is recalculated with the new bank line. At that time, the FOS of new bank will higher than 1, and the analysis process returns to Step 3 if the RWL is continually rising.

By building this framework (Figure 1), both processes of water level and soil erosion will be involved. Thus, the issues limiting previous research, such as the process of seepage, change of groundwater level, pore pressure, and unsaturated soil properties, are addressed in this study.

The process for determining the FOS in SLOPE/W using the limit equilibrium method (LE method). Calculations for the FOS are performed by dividing the riverbank block into vertical slices. The forces acting on a slice are shown in Figure 6 for a composite slip surface [45].

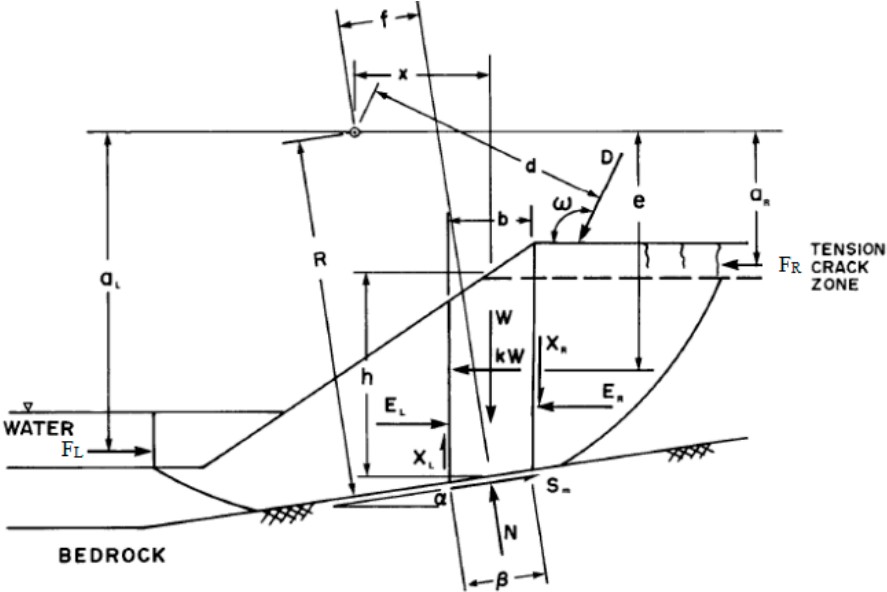

**Figure 6.** Forces acting on a slice of a sliding mass with a composite slip surface [45].

In Figure 6, $W$ is the total weight of a slice of width $b$ and height $h$; $N$ is the total normal force on the base of the slice; $S_m$ is the force mobilized on the base of each slice; $E_L$ and $E_R$ are the horizontal interslice normal forces at the left and right sides of the slice, respectively; $X_L$ and $X_R$ are the vertical interslice shear forces at the left and right sides of the slice, respectively; $D$ is an external point load; $kW$ is the horizontal seismic load; $R$ is the radius for a circular slip surface; $\alpha$ is the angle between the

tangent to the center of the base of each slice and the horizontal; $\beta$ is the sloping distance across the base of a slice; $\omega$ is the angle of the point load from the horizontal; $f$ is the perpendicular offset of the normal force from the center of rotation; $x$ is the horizontal distance from the centerline of each slice to the center of rotation; $h$ is the vertical distance from the center of the base of each slice to the uppermost line in the geometry; $e$ is the vertical distance from the centroid of each slice to the center of rotation; $d$ is the perpendicular distance from a point load to the center of rotation; $a$ is the perpendicular distance from the resultant external water force to the center of rotation; and $F$ is the resultant external water force $F_{L,R}$ is the resultant external water force at left and right sides of failure block.

The FOS can be computed based on moment equilibrium ($F_m$) and horizontal force equilibrium ($F_f$). These factors of safety may vary depending on the percentage ($\lambda$) of the force function used in the computation. In this study, the FOS was computed by using the Spencer method, because this method satisfied both moment equilibrium and horizontal force equilibrium. As shown in Figure 7, FOS is the crossover point of the moment and force equilibrium FOS values. The following equations of statics in solving for the FOS by $F_m$ and $F_f$:

$$FOS = F_m = \frac{\sum\left[c'\beta R + \left\{N - u_w\beta\frac{\tan\varphi^b}{\tan\varphi'} - u_a\beta\left(1 - \frac{\tan\varphi^b}{\tan\varphi'}\right)\right\}R\tan\varphi'\right]}{\sum Wx \pm \sum Fa - \sum Nf} \tag{15}$$

$$FOS = F_f = \frac{\sum\left[c'\beta\cos\alpha + \left\{N - u_w\beta\frac{\tan\varphi^b}{\tan\varphi'} - u_a\beta\left(1 - \frac{\tan\varphi^b}{\tan\varphi'}\right)\right\}\tan\varphi'\cos\alpha\right]}{\sum N\sin\alpha \pm \sum F} \tag{16}$$

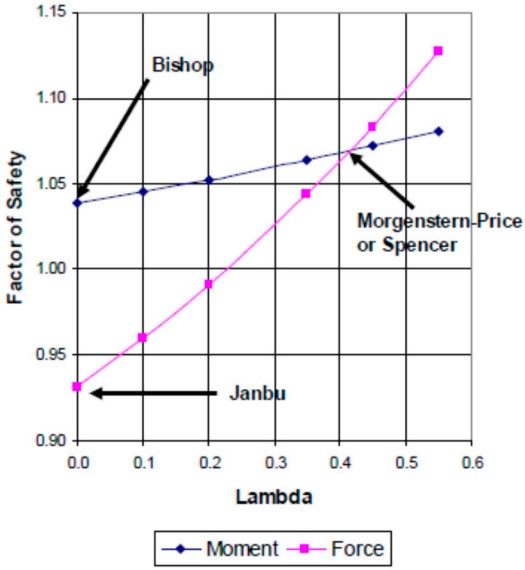

**Figure 7.** Factor of Safety (FOS) versus lambda plot.

## 3. Study Area Conditions

Figure 8 shows the study area of the Red River bank, part of the Red River delta in Hanoi city in the north of Vietnam. Ngoc Thuy (NT1), Xuan Canh (XC) and Hai Boi (HB) are natural bank areas and have been strongly eroded by water flow in the bending area. The riverbank in the Xuan Canh area was selected for simulating in the analysis model. This model includes the effects of both seepage and soil erosion by water shear stress near the riverbank. Three broad categories of data are required to run these simulation models: (1) bank geometry and stratigraphy data, (2) hydraulic boundary, and (3) geotechnical data.

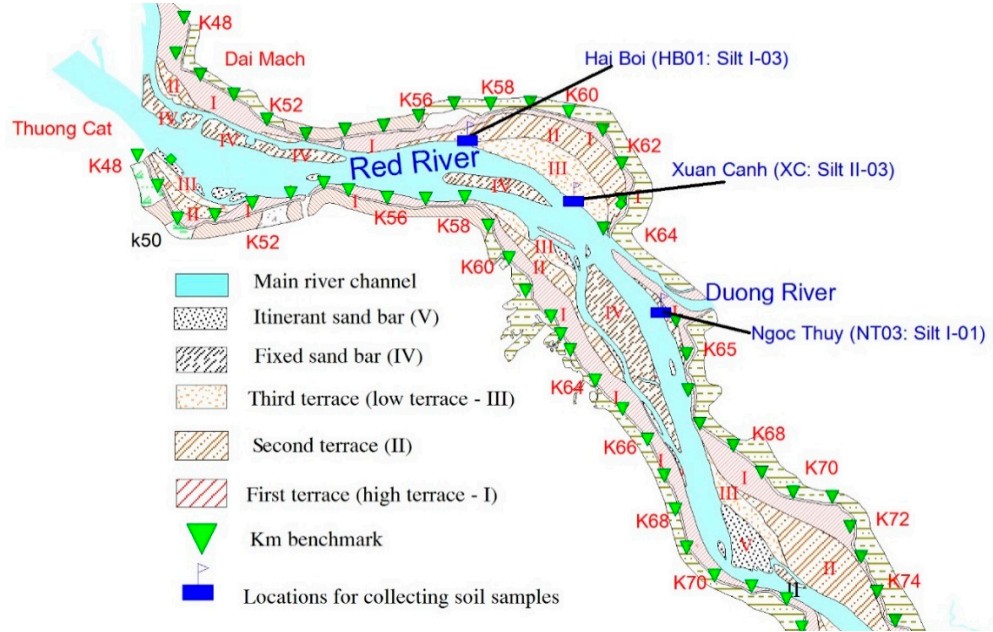

**Figure 8.** The study area.

The geometric behaviors of riverbanks and the hydraulic boundary are listed in Table 1. This riverbank has a bank surface elevation of 10 m, with a bank slope of 81°. Hydraulic dynamics include the initial RWL, the initial pore pressure, the change in RWL at different rates during rising and drawdown, and the distribution of shear stress near the bank, which causes soil erosion. The initial RWL, set at 0 m, is the elevation of RWL before the rainy season. The initial pore water pressure is defined as the maximum negative pressure head at 7 m so that the negative pressure at the bank surface is −50 kPa (Figure 9). A simulated flood event with a rate of RWL change of 1 m/d for both rise and drawdown is used for these models. These same data are used to determine the water shear stress profile.

**Table 1.** Bank soil layers and hydraulic conditions used in simulations of riverbank stability.

| Riverbank Properties | Unit | Descriptions |
|---|---|---|
| Bank surface elevation (H) | (m) | 10 |
| Bank slope | (°) | 81 |
| Soil layer and elevation of surface soil layer | (m) | Layer 1: from 10 m to (−1) m soil bank<br>Layer 2: from (−1) m down bed sand |
| Initial elevation of RWL | (m) | 0 |
| Initial pore pressure (Maximum negative head) | (m) | 7 m |
| Rate of water level change | (m/d) | 1 |
| River slope | (m/m) | 0.0001 |

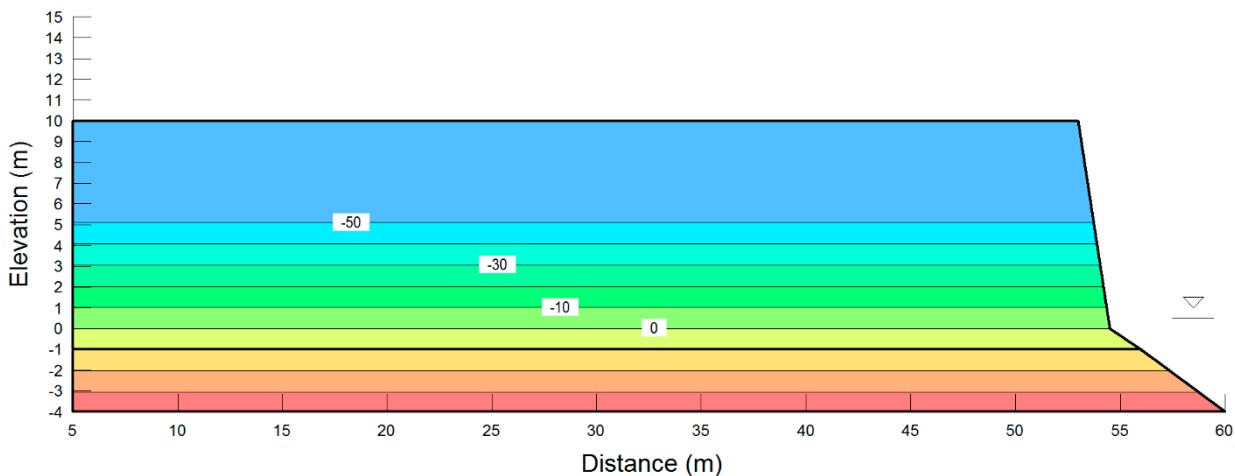

**Figure 9.** Configuration of riverbank.

To simulate riverbank stability and erosion, three different soil materials (Silt I-01 (NT03), Silt I-03 (HB21), and Silt II-03 (XC)) will be set up in the same riverbank configuration as Xuan Canh riverbank. Table 2 shows the properties of these three soils. The stability of the riverbank was then analyzed in the following two scenarios: Case 1—The soil bank is eroded based on the distribution of shear stress with depth. In this case, the soil bank is eroded, and then the bank geometry is updated as described in Section 2.3. Case 2—There is no soil erosion, and bank geometry remains constant as the RWL changes at a rate of 1 m/d. Figure 1 shows different steps in both analysis processes with erosion and without erosion.

**Table 2.** Soil properties used in riverbank stability analysis.

| Soil Properties | | Silt I-01 (NT 03) | Silt I-03 (HB21) | Silt II-03 (XC) |
|---|---|---|---|---|
| Sand | 0.25–0.075 mm | 5 | 15 | 35 |
| Silt | 0.075–0.005 mm | 70 | 70 | 55 |
| Clay | <0.005 mm | 25 | 15 | 10 |
| Saturated volumetric water content (%) | | 30 | 42 | 45 |
| Air-entry value (kPa) | | 27 | 20 | 10 |
| Hydraulic conductivity (m/s) | | $4.32 \times 10^{-7}$ | $7.39 \times 10^{-7}$ | $2.24 \times 10^{-6}$ |
| Cohesion force (kPa) | | 7.5 | 5.0 | 5.0 |
| Internal friction angle (°) | | 32 | 32 | 30 |
| Critical erosion shear stresses. (kPa) | | 1.5 | 1.1 | 1.0 |

Comparing the results of these two scenarios allows the stability of the riverbank to be evaluated using the calculated FOS. Cases of erosion of the bed sand layer were not considered; riverbank stability was only assessed by tracking changes in the silt bank layer.

## 4. Results and Discussion

### 4.1. Effects of Soil Erosion to Riverbank Stability

Figure 10 shows the soil erosion curves for three different soils of Silt I-01 (NT03), Silt I-03 (HB21), and Silt II-03 (XC), which have increasing sand contents of 5%, 15% and 36%, respectively. The critical shear stresses, defined as the shear stress, are small and are nearly consistent between all soils of these groups; however, this value increases significantly with increasing fine grain size content.

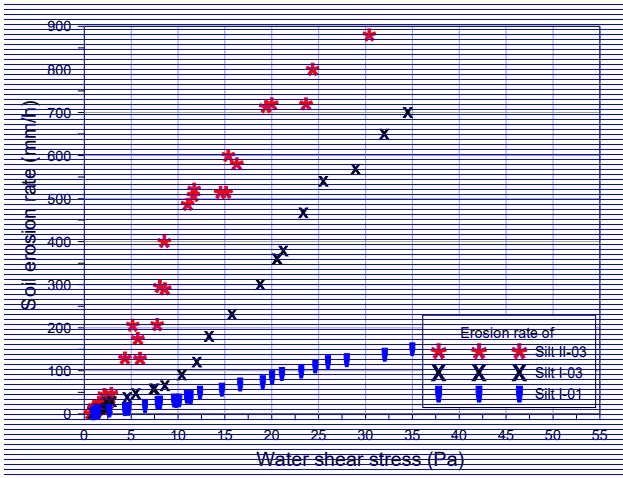

**Figure 10.** Erosion rates of Silt I-01, Silt I-03 and Silt II-03 at a density of 15 kN/m$^3$.

Figure 11 (left) shows the calculated shear stress ranging from the initial RWL at 0 m to the peak RWL at 10 m. The shear stress is calculated by Equation (10), with $D$ = 10 m, and y changing from 0 to 10 m. The shear stress increases with the decrease of y and reaches a maximum in the bank toe at 0 m, thus demonstrating that the bank toe is often the most heavily eroded region of the riverbank. From this shear stress profile, the responding erosion rates for Silt I-01, Silt I-03, and Silt II-03 soils are determined by the soil erosion rate curve in Figure 10. Then the soil erosion width for Silt I-01, Silt I-03, and Silt II-03 riverbanks are calculated using Equation (13). Figure 11 (right) shows the soil erosion width responding to the shear stress of these three soils. The soil with higher erosion rate has a larger erosion width; these results may have a great effect on the riverbank stability. These are the results of the shear stress and erosion width corresponding to the RWL rising 10m. Analysis of the difference of RWL used the same method and process.

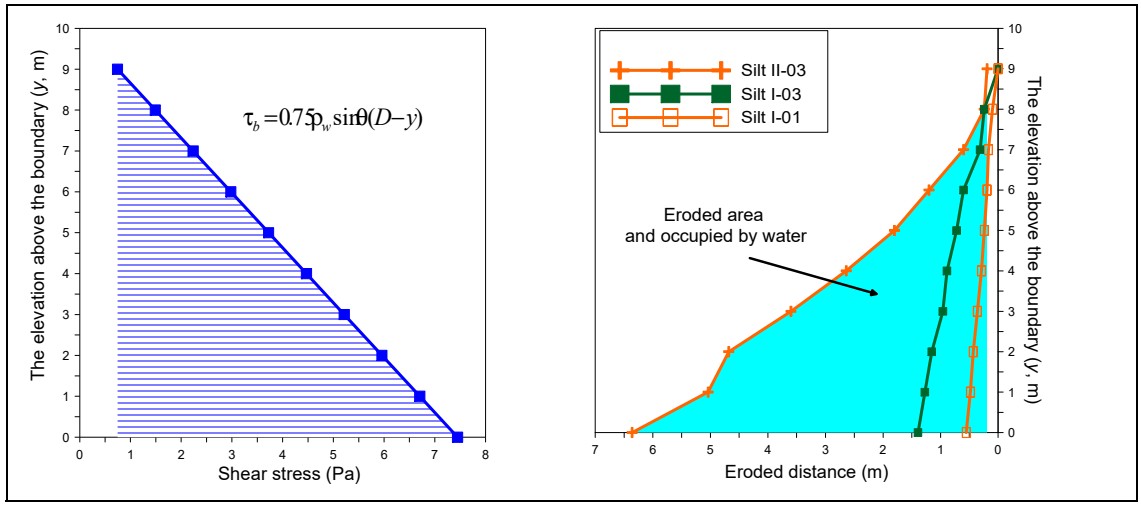

**Figure 11.** Shear stress boundary near riverbank (**left**), and eroded distance (**right**).

To simulate changes in the geometry of the riverbank due to bank soil erosion, this simulation model was run over multiple periods. During the first period, from Days 1–4, the RWL rose from 0 m on Day 1 to 3 m on Day 4; during this time, the soil was insignificantly eroded, and the bank geometry did not change substantially. The bank line was updated after the last day of the first period (on the fourth day). The stability of the riverbank after this first period was determined by calculating the FOS when the RWL was at 3 m, and the bank line was eroded by three days of flow. After this first period,

the next stage was divided into a series of day-long intervals. A one-day interval was chosen because the riverbank may have failed after only a short time of soil bank erosion. At the end of each day, the bank line was updated and the new FOS was calculated.

Figures 12 and 13 show results of the FOS corresponding to RWL increasing and soil erosion of the three soils. Figure 12 includes the change of RWL, and the result of FOS with both change of RWL and soil erosion, and the results of FOS with only change of RWL for soil Silt II-03. In the case without soil erosion, the change of FOS has the same trend as the changes of RWL, the FOS increases with increasing RWL, and FOS decreases with decreasing RWL. These trends happen for all three soils. However, the FOS changes with a different trend in the case of both RWL changes and soil erosion as shown in Figure 13 and outlined in the following description.

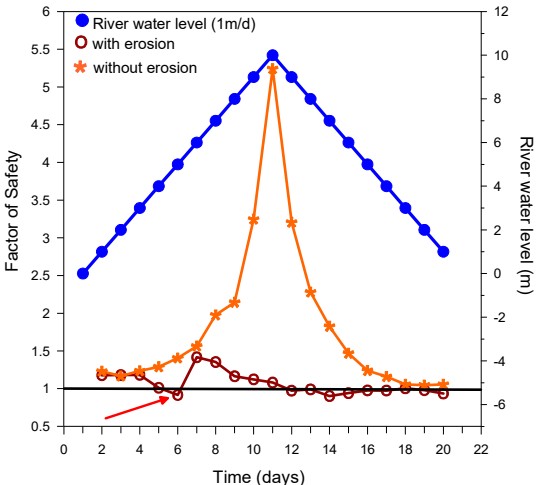

**Figure 12.** Evolution of bank stability for Silt II-03.

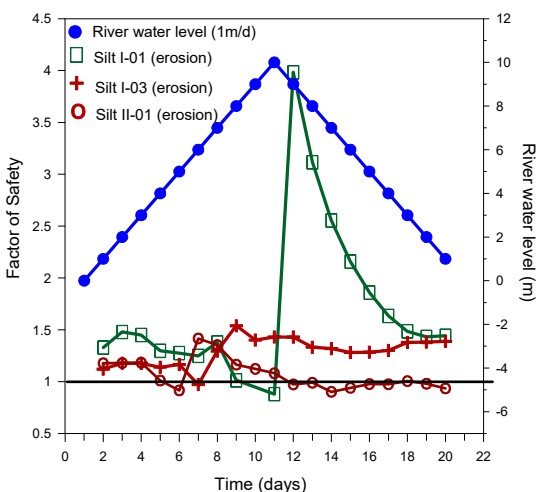

**Figure 13.** Comparison of bank stability with erosion for three soils.

In the first period, 0–3 m, in which the shear stress was small, the FOS increased slightly. The FOS had already begun to decrease slightly during the last two days of the first period when the water shear stress exceeded the critical shear stress. Following the first period, the FOS continued to decrease. The bank line above RWL remained the same at all steps until the FOS exceeded 1, but the bank line under the RWL was updated after this rise in RWL to reflect the cumulative erosion width. In the riverbank of Silt II-03 soil, the FOS decreased to 0.916 when the RWL rose to 5 m after six days. This bank collapsed one day earlier than the riverbank of Silt I-03, in which the FOS value decreased to

0.972 at 6 m. The riverbank of Silt I-01 collapsed last when the RWL rose to 10 m after 11 days, and the FOS dropped to 0.88. Generally, the FOS decreased to a value of less than 1 first in the Silt II-03 riverbank, then in the Silt I-03 riverbank and finally in the Silt I-01 riverbank (Figure 13).

After riverbank failure, the bank line was modified based on the failure surface, as the riverbank was assumed to have fallen into the river channel along the failure surface. Consequently, the new bank line above the current RWL represented the failure surface, and the bank line under the current RWL represented the bank line updated to reflect the occurrence of soil erosion. The new bank line was updated, and the pore pressure and RWL were inherited from the previous period, producing a new bank with higher stability. Therefore, the FOS suddenly increased after these bank failures, and the maximum FOS values were obtained in this step during the simulated flood event. The maximum value was obtained using the updated bank geometry resulting from failure by erosion, in which the resulting FOS values were 3.98 for Silt I-01, 1.56 for Silt I-03, and 1.42 for Silt II-03. Following this point, as the RWL continued to rise, the bank soil was eroded at a higher width, thus decreasing the FOS values.

Figures 14–16 show the evolution of riverbank stability with erosion at the bank toe for the three types of modeled soil bank materials. There are three riverbank stability states: (1) at the end of the first state, after the RWL rose from 0 m to 3 m; (2) when the FOS dropped to a value of less than 1 and the original bank collapsed; and (3) after the bank line was updated and the riverbank state recovered. These figures provide details of the hydraulic and geometric conditions corresponding to the phases discussed above. Changes in RWL are shown on the riverside, and changes in the groundwater table (dashed blue line) and soil erosion are shown on the bank side. The soil area under the RWL was eroded and removed, after which this entire area was occupied by river water.

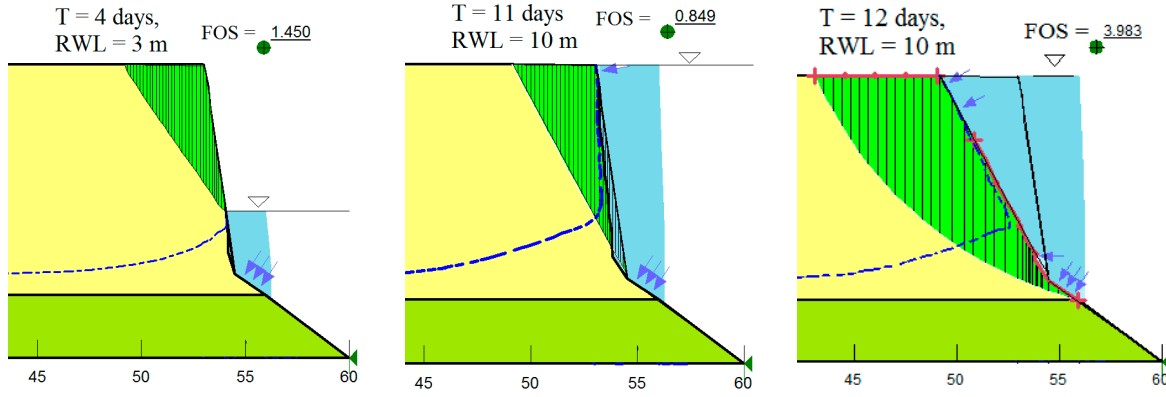

**Figure 14.** Simulated Silt I-01 riverbank after rise in RWL to 3 m, 10 m (before bank update), and 10 m (after bank update).

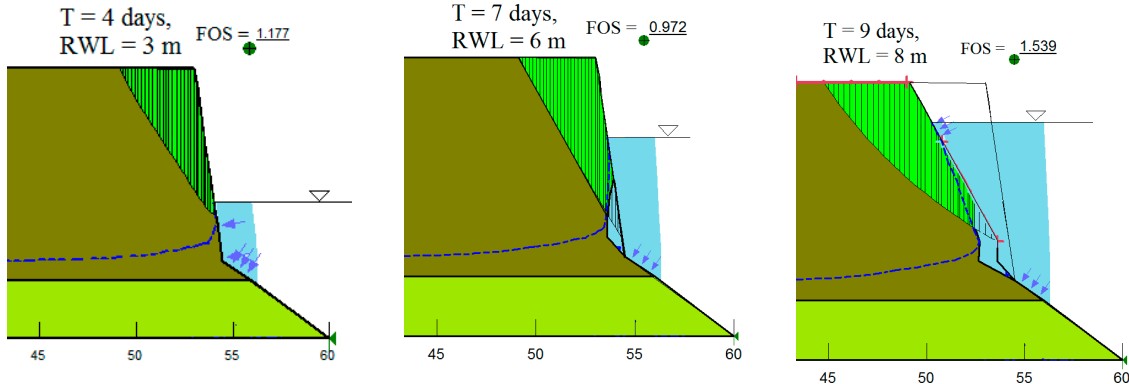

**Figure 15.** Simulated Silt I-03 riverbank after rise in RWL to 3, 6 and 8 m.

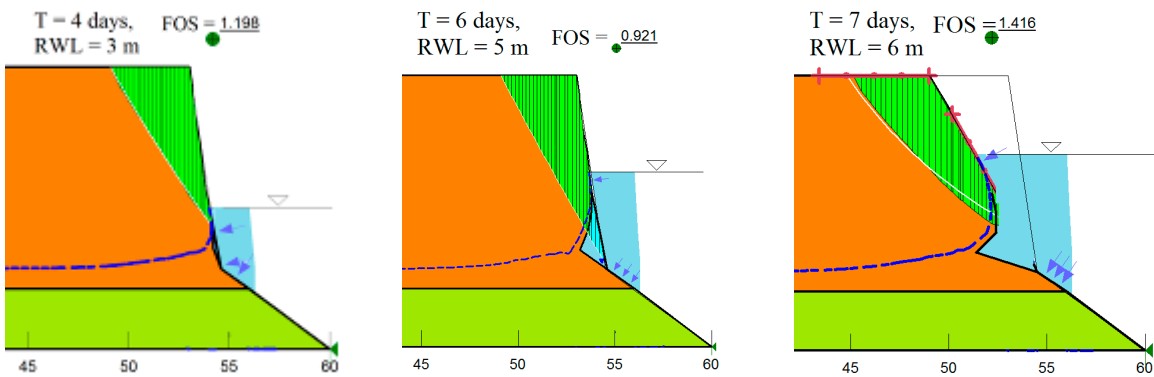

**Figure 16.** Simulated Silt II-03 riverbank after rise in RWL to 3, 5 and 6 m.

The soil erosion and riverbank failure process depend on the erosional behavior of the bank soil. Three types of soil were modeled; in order of increasing erosion rate, these are Silt I-01, Silt I-03, and Silt II-03, which have critical shear stresses of 1.49, 1.09, and 1.00 Pa, respectively. Critical shear stresses do not vary substantially among these soils, but their erosion rates vary quite substantially at higher shear stresses. Less cohesive soils with higher erosion rates become unstable earlier. This can be seen in the recorded times to failure determined in the simulations, which are six days for Silt II-03, seven days for Silt I-03, and 11 days for Silt I-01. The less cohesive soil bank (Silt II-03) fails at lower RWL (6 m) and more rapidly (six days) than the more cohesive soil banks (Silt I-03 and Silt I-01). Moreover, different soils produce different riverbank failure shapes. When riverbank failure occurs, the overhang shape is greater in the less cohesive soil (Silt II-03) than in the more cohesive soil (Silt I-01) because of the higher width of soil erosion at the toe of the riverbank, then the riverbank in the less cohesive soil (Silt II-03) fails earlier than that in cohesive soil.

The effects of soil properties on the process of riverbank failure have previously been discussed in [19,20,22–25,30], which simulated this process at the same river water level. The riverbank in non-cohesive soil banks (mainly those comprising sandy soil) tends to fail near the water surface, whereas cohesive bank failure occurs near the bank toe, where the water velocity is much higher [23]. A series of simulations modeling the full processes of riverbank failure, depicting the regions both under and above the undercutting overhang were performed in [19,20]. These results demonstrate that soils with higher soil density have higher resistance force and thus higher undermining failure depths at which the riverbank failure occurs. These previous studies clearly illustrated the effects of soil properties on riverbank failure in cases without changes in the RWL. In this study, modeled simulations were performed using variations in RWL and soils. The Silt I-01 bank failed at an RWL of 10 m and a maximum erosion width of 0.55 m; the Silt I-03 bank failed at an RWL of 6 m and a maximum erosion width of 0.72 m; and the Silt II-03 bank failed at an RWL of 5 m and a maximum erosion width of 1.2 m. During flood events in which RWL increases at the same rate, the less cohesive soil bank will be eroded in a larger toe cantilever shape, and riverbank failure will occur earlier than the more cohesive soil banks. The results obtained in this study include the effects of soil properties and RWL conditions and show that both soil bank properties and RWL can greatly affect riverbank stability. The upper riverbank overhang failure may occur as a type of topping failure, as described in [19,20].

### 4.2. Complexity of Factors in Riverbank Stability Analysis

In the scenario analyzed without soil erosion, the results demonstrate that FOS increased during RWL rise and decreased during drawdown. The studies analyzing the type of mass failure [2–9] illustrated the effects of factors, such as confining pressure, pore pressure, and unsaturated soil properties on riverbank stability. In particular, Duong et al., (2014) [6] clarified cases of riverbanks controlled by soil hydraulic conductivity. In cases of low hydraulic conductivity (<$10^{-6}$ m/s) and those in which soil erosion does not occur in the toe of the riverbank, the confining pressure is the

main factor affecting the FOS value. However, when riverbank soil has high hydraulic conductivity ($>10^{-6}$ m/s), increasing the RWL will cause the pore water pressure to also increase. In soils with higher hydraulic conductivity, pore water pressure and the rate of water level change are the primary factors affecting the FOS value. Those results may contrast with results obtained in this study, as well as those obtained from studies analyzing the effects of soil erosion and RWL on riverbank stability [17,18,21]. As the rates of RWL change increase, water stress increases more rapidly, and the soil is more eroded. Because both higher water shear stress and higher rates of soil erosion produce larger soil erosion widths, the bank failures are caused more rapidly. In this study, RWL change was defined as 1 m/day and erosion in the toe of bank was accounted for. Riverbank failure occurred after the RWL rose by 5 m, 6 m, and 10 m and after six days, seven days, and 11 days for the soil banks of Silt II-03, Silt I-03, Silt I-01, respectively. However, without accounting for soil bank erosion, the riverbank of Silt II-03 only failed when its hydraulic conductivity was higher than the increase in the RWL [6]. Combining these results with those of previous studies, the understanding of the mechanisms of riverbank failure should cover all factors, such as the rate of RWL fluctuation, seepage and soil erosion. With low rates of RWL change (especially when they are equal to or smaller than the hydraulic conductivity of the soil), riverbank failure may occur as a mass failure even without toe-bank soil erosion due to increasing pore pressure by seepage process. With high rates of RWL change as well as high water stress, toe-bank soil erosion will occur, and the riverbank fails as an overhang riverbank failure. In the latter case, the soil properties and erosion rate are the main factors causing riverbank failure, and the confining pressure continually supports the riverbank when the RWL rises. The pore water pressure and soil suction may not significantly affect riverbank stability because the groundwater in the riverbank area is not greatly increased by the increase of the RWL.

## 5. Conclusions

This paper simulated the processes of riverbank failure using three different types of bank soil and by modeling both mass failure and undercutting erosion. By combining the results of these simulations with those of previous studies, the following conclusions can be reached.

Soil erosion is closely related to the content of fine-grained particles. The higher the fine content, the lower the erosion rate. The silt soils found along the Red River bank have a very high erosion rate. Their critical shear stress is small and is nearly constant at 1–1.5 Pa for a soil density of 15 kN/m$^3$.

During RWL rise, the soil bank erosion width and overhang shape develops due to increased shear stress. In the same configuration of the riverbank and changing RWL, the bank has a higher soil erosion rate, and riverbank failure occurs within a shorter time and at a lower RWL. In the three cases of Silt I-01, Silt I-03, and Silt II-03, which are listed in order of increasing erosion rate, riverbank failure occurs after 10 days, seven days and six days, respectively. The less cohesive soil bank (Silt II-03) fails at a lower RWL (6 m) and in a shorter amount of time (six days) than the more cohesive soil banks (Silt I-03 and Silt I-01).

The shape of the riverbank cantilever failure depends on the soil properties and RWL. Without changing the RWL, riverbanks with higher resistance force (i.e., higher cohesive force and soil density) experience failure with larger and deeper overhang erosion. In flood events in which RWL increases at the same rate, less cohesive soil banks will be eroded in larger toe overhang shapes than more cohesive soil banks, and riverbank failure will occur earlier.

In cases with a low RWL rising rate (especially when equal to or smaller than soil hydraulic conductivity), riverbank failure may occur as a mass failure. In cases of both high RWL rising rate and high water stress, toe-bank soil erosion will occur, and the riverbank will undergo overhang riverbank failure. The soil properties and erosion rate are the main factors causing riverbank failure.

**Author Contributions:** Conceptualization, T.D.T. and D.D.M.; methodology, T.D.T. and D.D.M.; software, T.D.T.; validation, T.D.T. and D.D.M.; formal analysis, T.D.T.; investigation, T.D.T. and D.D.M.; resources, T.D.T.; data curation, T.D.T. and D.D.M.; writing—original draft preparation, T.D.T.; writing—review and editing, T.D.T. and D.D.M.; visualization, T.D.T.; supervision, D.D.M.; project administration, T.D.T.; funding acquisition, T.D.T.

**Funding:** This research was funded by the project Code 105.08-2015.24, which was sponsored by Nafosted, Ministry of Science and Technology, Vietnam.

**Acknowledgments:** This paper was completed with the support of the project Code 105.08-2015.24, which was sponsored by Nafosted, Ministry of Science and Technology, Vietnam. The authors express their sincere gratitude for this support.

**Conflicts of Interest:** The authors declare no conflicts of interest.

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
