# Peer review of "Riverbank Stability Assessment under River Water Level Changes and Hydraulic Erosion"

_water, doi:10.3390/w11122598_

Round 1

Reviewer 1 Report

line 64 - language

Fig. 6- legend doesn't match the map

Fig. 10 and Fig. 11 - the way of curves description should be unified in both figures

chapter Discussion should be separated from results or title of the chapter 4 should be changed for "Results and Discussion"

Author Response

Dear Reviewer 

I appreciate your recommendations which greatly support me to improve my manuscript.

Please find my response to all your recommendation in attack file 

Thank you very much 

Duong Toan 

Reviewer 2 Report

The manuscript Riverbank Stability Assessment under River Water Level Changes and Hydraulic Erosion reports on a method to simulate the collapse of riverbanks and its related factor of safety. This combines laboratory tests, field evidences and mathematical simulations, therefore the research work appears relevant.

While two mechanisms are clearly identified that are mass failure and undercutting erosion, the content of the manuscript mainly focus on the latter. For sake of clarity, some details must be given on mass failure too, although the topic is well consolidated in the literature and I suppose that authors consider the accounting for undercutting erosion as the novelty of their contribute. However the Results section discuss the performed simulations which account for both mechanisms which then should support the conclusion remarks. Therefore, in the methodological part, the role of pore water pressure in determining a mass failure should be briefly recalled. Related to that, the assessment of safety factor must be explained since a reader may be not know GeoSlope software (myself neither) and for what I know a variety of approach exist for FOS assessment which account for a variety of mechanisms. In other words, I think that SEEP/W and SLOPE/W should be not take for granted. Generally speaking, I believe that a scientific contribute should be complete and exhaustive in terms of involved fundamental concepts and regarding that it should not reference to existing software.

I found that the use of technical/scientific vocabulary is not always accurate, this is also related to an uneven of English but I’m not native English speaking, therefore my review on this is incomplete. For example from page 6 of 19, it can be read: “Equation 9 shows the distribution of shear stress from the riverbed to the surface of the channel, where D is the depth from the current RWL surface to the riverbed or the initial RWL, y is the depth from the riverbed or the initial RWL to a shear stress point, and the shear depth d is equal to (D−y).”

The “surface of the channel” should be the “water surface” “D is the depth from the current RWL surface to the riverbed or the initial RWL” should be simply “D is the current water depth at thalweg” “y is the depth from the riverbed or the initial RWL to a shear stress point” should be “y is the elevation above the thalweg of a considered section across the channel” “the shear depth d is equal to (D−y)” should be “the water depth d at considered section is (D-y)”

Another not appropriate frequently used wording is “erosion distance” that should be “cumulated erosion”, “erosion depth”, “total erosion”, “erosion width”. Many others examples can be found, therefore I recommend a careful rewriting.

Regarding the scientific/technical contents I have some major concerns. To be honest I must stress that these concerns may have been biased from the mentioned not proficient use of English.

In the methodological part, a laboratory test is presented to eventually assess the erosion rate and corresponding shear stress for a soil sample. While the setup appears appropriate to eventually produce erosion rate under controlled conditions, the shear stress assessment by means of the friction coefficient “f” refers to experimental evidences on smooth pipes mainly achieved between the end of ninth century and the seventies of the last century. I think that, for the proposed tests, the soil sample is not subjected to the shear stress of smooth pipe used in the setup. In fact I would expect a change in the water surface slope and depth at the sample section along the setup pipe. Again in the methodological part, equation from 7 to 10 refer to the shear stress at river/channel bed in case of uniform flow, that derives from the momentum balance application to a river channel straight reach. In this case, the streamflow is aligned to the channel and the water surface is streamwise slopped. Given this oversimplified geometry, the shear stress is proportional to water depth, density and slope, the latter also corresponds to bed slope in case of uniform flow only. I do not understand if in equations from 7 to 10 and following considerations/asseement/modelling/results, the sin of teta is intended to be the streamwise slope of the channel (i.e., the water surface mean slope of the considered river reach). Apparently, in the proposed manuscript, sin(teta) is the bank slope of cross section that hardly confuses my understanding! Related to shear stress assessment, the Figure 6 presents a typical river morphology with bends, bars and bifurcations. In this case the assessment of shear stress by using the momentum balance result (i.e., equations from 7 to 10) is clearly an oversimplification that call for justification/clarification. In this case study, locally, for example at river bend, the shear stress distribution may be not univocally related to water depth that is typically reflected in a variation of the water slope across the channel. This may contradict the trivial finding that shear stress linearly scales with water depth as reported in Figure 9 - left panel. By the way, the y-axis should be “bed level” instead of “water level”, i.e. maximum bed level “y”, corresponds to minimum water depth “D-y” and therefore minimum shear stress in a momentum balance result fashion.

Overall I see the potential of performed work but the text appears far from being ready for publication and results/conclusions appear naive since the actual complexity of river morphodynamics is fairly considered/discussed.

Author Response

Dear Reviewer 

I appreciate your recommendations which greatly support me to improve my manuscript.

And I sorry to respond your recommendations late because I need more time to understand and build carefully responding to all your recommendations.

Please see detail response in the attach file

Thank you very much 

Yours sincerely, 

Duong Toan

Round 2

Reviewer 2 Report

The text was significantly improved and I recognize the effort made in answering to my concerns. Still, my opinion is that the method applied to assess shear stress in the laboratory tests is questionable although practical. However I can live with that, most likely in future works, the assumption of smooth pipe should be relaxed and the actual shear stress above the terrain sample will be more properly evaluated (e.g., turbulence fluctuation measurement). 

Equation 4 should be D=2ab/(a+2b)

Please give a justification of the using of 0.75 in Equation 10, I guess this is because of the position of depth averaged velocity (and shear stress) along the water column but I do not get why the maximum is not taken (i.e., tao0 at bed); by the way, why this coefficient changed from 0.75 to 0.76 in Figure 11 left panel? 

Author Response

Dear Reviewer 

Thank you very much for your approval of my responding and continue to support me improve my paper.

Please find my responding to all your recommendations in Round 2 in the attach file.

Yours sincerely,  

Duong Thi Toan

This manuscript is a resubmission of an earlier submission. The following is a list of the peer review reports and author responses from that submission.